# *Treponema denticola* Induces Neuronal Apoptosis by Promoting Amyloid-β Accumulation in Mice

**DOI:** 10.3390/pathogens11101150

**Published:** 2022-10-05

**Authors:** Linrui Wu, Xinyi Su, Zhiqun Tang, Lixiang Jian, He Zhu, Xingqun Cheng, Hongkun Wu

**Affiliations:** 1State Key Laboratory of Oral Diseases, National Clinical Research Center for Oral Diseases, Department of Geriatric Dentistry, West China Hospital of Stomatology, Sichuan University, Chengdu 610041, China; 2Department of Stomatology, Beijing Hospital, National Center of Gerontology, Institute of Geriatric Medicine, Chinese Academy of Medical Sciences, Beijing 100730, China; 3Department of Oral Mucosal Diseases, Affiliated Stomatology Hospital of Guangzhou Medical University, Guangzhou Key Laboratory of Basic and Applied Research of Oral Regenerative Medicine, Guangzhou 510013, China

**Keywords:** *Treponema denticola*, Alzheimer’s disease, chronic periodontitis, neuronal apoptosis, amyloid-β accumulation, *Porphyromonas gingivalis*

## Abstract

**Background:** Neuronal apoptosis is a major contributor to Alzheimer’s disease (AD). Periodontitis is a significant risk factor for AD. The periodontal pathogens *Porphyromonas gingivalis* and *Treponema denticola* have been shown to initiate the hallmark pathologies and behavioral symptoms of AD. Studies have found that *T. denticola* infection induced Tau hyperphosphorylation and amyloid β accumulation in the hippocampi of mice. Aβ accumulation is closely associated with neuronal apoptosis. However, the roles of *T. denticola* in neuronal apoptosis remain unclear and its roles in AD pathology need further study. **Objective:** This study aimed to investigate whether oral infection with *T. denticola* induced alveolar bone loss and neuronal apoptosis in mice. **Methods:** C57BL/6 mice were orally administered with *T. denticola*, Micro-CT was employed to assess the alveolar bone resorption. Western blotting, quantitative PCR, and TUNEL staining were utilized to detect the apoptosis-associated changes in mouse hippocampi. N2a were co-cultured with *T. denticola* to verify in vivo results. **Results:** Mice infected with *T. denticola* exhibited more alveolar bone loss compared with the control mice. *T. denticola* oral infection induced neuronal apoptosis in hippocampi of mice. Consistent results of the apoptosis-associated protein expression were observed in N2a cells treated with *T. denticola* and Aβ_1–42_ in vitro. However, the Aβ inhibitor reversed these results, suggesting that Aβ_1–42_ mediates *T. denticola* infection-induced neuronal apoptosis. **Conclusions:** This study found that oral infected *T. denticola* caused alveolar bone loss, and induced neuronal apoptosis by promoting Aβ accumulation in mice, providing evidence for the link between periodontitis and AD.

## 1. Introduction

Alzheimer’s disease (AD) is a progressive and irreversible neurodegenerative disorder characterized by memory and cognitive decline, disorientation, and personality changes [1]. It is a chronic disease with a high incidence of 10–50% in people over age 65, affects 43.8 million people worldwide, and is the fifth leading cause of death in the world [2,3]. Senile plaques composed of Aβ, neurofibrillary tangles (NFTs) formed by hyperphosphorylated tau, and progressive loss of synapses and neurons [4] are pathological hallmarks of AD. The causes of AD are unclear. However, patients with AD exhibit neuroinflammation associated with infection, including microglial cell activation and altered inflammatory factor profiles, implying that infection may be an etiology of AD [5].

Chronic periodontitis is a common and widespread oral infectious disease [6]. Periodontal pathogens and their virulence factors, such as *Porphyromonas gingivalis* (*P. gingivalis*) and *Treponema denticola* (*T. denticola*), can enter distant organs through the circulatory system via erosive and swollen periodontal tissue, resulting in the onset and progression of a variety of systemic diseases [7]. Chronic periodontitis has been identified as a major risk factor for AD [8,9,10]. Epidemiological investigations have shown that the degree of cognitive impairment in patients with severe periodontitis is three times greater than that of patients with mild periodontitis or without periodontitis [11]. In the elderly with normal cognitive function, alveolar bone resorption is positively correlated with Aβ deposition in brain tissues [12]. Studies have shown that *P. gingivalis* can enter the brain tissue of AD patients, and induce Aβ accumulation [13,14], tau hyperphosphorylation, neuroinflammation, and neuron loss [15,16]. Furthermore, as a predominant spirochete in the subgingival plaque of the gingival crevice and periodontal pocket, a possible link between *T. denticola* and AD has been reported [17]. One study demonstrated that spirochetes were found in 91 percent of 495 brain and blood samples from AD patients, but 0 percent of 185 samples from controls [18]. *T. denticola* has also been detected in the postmortem brain tissues of AD patients using PCR [19]. Our previous research demonstrated that oral infections of *T. denticola* could promote AD pathology in the hippocampi of mice, including an increase in Aβ burden [20], tau hyperphosphorylation, and neuroinflammation [21]. However, further studies are needed to determine the relationship between *T. denticola* and AD. 

Neuronal apoptosis is a major contributor to neurodegenerative disorders such as AD, Parkinson’s disease, and amyotrophic lateral sclerosis [22]. Johnson reported on the potential role of neuronal apoptosis in AD in 1994 [23]. Numerous studies have found the relationship between Aβ and neuronal apoptosis in AD [24,25]. Aβ accumulation, in particular, contributes to neurodegeneration by activating pro-apoptotic proteins to induce mitochondrial dysfunction [26]. Zhao and Huang et al. discovered that intracellular Aβ aggregation occurred in the early stages of AD, and Aβ aggregation in the cytoplasm might cause structural damage in synapses, functional abnormalities, and neuronal apoptosis [27]. Kawahara reported that Aβ oligomers caused neuronal cell death in the later stage of AD [28]. The most common subtypes of Aβ, Aβ_1–40,_ and Aβ_1–42_ are neurotoxic and play key roles in neuronal apoptosis and cognitive impairment in AD [29]. *T. denticola* infection induced Aβ_1–40_ and Aβ_1–42_ accumulation in mouse hippocampi in our previous study [20]. Therefore, we hypothesized that *T. denticola* infection might induce neuronal apoptosis and accelerate the pathological progression of AD by increasing the Aβ burden.

In this study, a mouse model was used to investigate whether oral infection with *T. denticola* induced alveolar bone resorption and neuronal apoptosis in the hippocampi. Mouse neuroblastoma N2a cells were incubated with *T. denticola* suspension in vitro to validate the pro-apoptotic effect and underlying mechanisms.

## 2. Results

### 2.1. T. denticola Induced Alveolar Bone Resorption and Neuronal Apoptosis in the Mouse Hippocampi 

*T. denticola* and *P. gingivalis* were detected in the saliva of all mice administered the specific bacteria orally, indicating that the bacteria colonized successfully (Appendix A). ABL was measured as the distance from the cementoenamel junction (CEJ) of the second maxillary molar to the alveolar bone crest (ABC) in three-dimensional reconstruction images. The Micro-CT images revealed that mice infected with *T. denticola* and *P. gingivalis* exhibited significantly more ABL than control mice (Figure 1A). TUNEL staining results revealed that hippocampi of mice in the *T. denticola* infection group and the *P. gingivalis* infection group exhibited significantly higher apoptosis rates than those in the blank control group (Figure 1B). The expression of cleaved caspase-3 was determined by western blotting. As shown in Figure 1C, the level of cleaved caspase-3 was significantly higher in the hippocampi of the *T. denticola* and the *P. gingivalis* infection groups compared with the blank control group. 

### 2.2. T. denticola Oral Infection Regulated the Expressions of Apoptosis-Associated Genes and Proteins

The hippocampi of *T. denticola*- or *P. gingivalis*- infected mice had higher levels of BCL2-like 11 (*Bim*) and BCL2-associated X protein (*Bax*), and lower levels of B cell leukemia/lymphoma 2 (*Bcl2*) and *Bclw* compared with the sham group (Figure 2A). Furthermore, *Bclw* showed the greatest change. Gene expression levels of *Jnk2*, *Jnk3*, and *Smac* were examined, and the results revealed that the expression levels of *Jnk2* and *Smac* were significantly up-regulated (Figure 2A). Moreover, BCL-W protein was significantly down-regulated, whereas SAPK/JNK and SMAC were significantly up-regulated in the hippocampi of the *T. denticola*- or *P. gingivalis*-infected groups (Figure 2B).

### 2.3. Amyloid-β_1–42_ Mediated T. denticola Infection-Induced Neuronal Apoptosis

To test if *T. denticola* directly promoted neuronal apoptosis, N2a cells were cultured with *T. denticola.* The results showed that apoptosis and the protein level of cleaved caspase-3 were significantly higher in N2a cells co-cultured with *T. denticola* than those in the control group, which was consistent with the results in vivo (Figure 3A,B). To further explore the potential mechanisms, N2a cells were stimulated with Aβ_1–40,_ and Aβ_1–42_. The apoptosis rates were significantly increased in the Aβ_1–42_- and *T. denticola*-treated groups, whereas there was no statistical difference between the Aβ_1–40_ group and the control group (Figure 3A). The levels of cleaved caspase-3 of Aβ_1–42_- and *T. denticola*-treated groups were significantly higher than in the control group (Figure 3B). The expression levels of SAPK/JNK and SMAC proteins were increased, while the expression of BCL-W protein was decreased in both the Aβ_1–42-_ and *T. denticola*-treated groups (Figure 3C). The expression levels of cleaved caspase-3 and BCL-W changed after Aβ_1–40_ treatment, while the expression levels of SAPK/JNK and SMAC did not differ between the Aβ_1–40_ and control groups (Figure 3B,C). 

Moreover, the N2a cells were pretreated with an Aβ inhibitor KMI1303, then cocultured with *T. denticola*. The apoptosis rate was significantly decreased following KMI1303 treatment (Figure 4A). Compared with the *T. denticola*-treated group, the levels of cleaved caspase-3, SMAC, and SAPK/JNK were significantly decreased, and the level of BCL-W, was significantly increased in N2a cells pretreated with KMI1303 (Figure 4B). All the results suggest that Aβ_1–42_ mediates *T. denticola* infection-induced neuronal apoptosis.

## 3. Discussion

Chronic periodontitis has been identified as a significant risk factor for AD [12]. Infection with periodontal pathogens, such as *P. gingivalis* and *T. denticola*, promotes AD pathogenesis in the hippocampi of mice, including Aβ accumulation, tau hyperphosphorylation, and inflammatory responses [13,14,16]. Noguchi and Moore reported that *T. denticola* could infect the cerebral cortex and cause atrophic dementia, cortical atrophy, and local amyloidosis as early as 1913 [30]. Increasing numbers of studies have suggested that *T. denticola* may contribute to AD pathogenesis [17,31,32,33]. Our previous study found that oral *T. denticola* infection caused bacterial invasion in the brain and increased the Aβ burden in the hippocampi of mice. Aβ accumulation is closely associated with neuronal apoptosis, which is a key factor in the pathological process of AD [22]. In this study, we aimed to explore how *T. denticola* induces neuronal apoptosis in AD.

We found an increased neuronal apoptosis rate following *T. denticola* exposure in vivo and in vitro. Based on our previous study, *P. gingivalis* infection was used as a positive control for apoptosis in mouse hippocampi. Our findings were consistent with previous research that found apoptotic changes in AD brain tissue slices, with apoptosis rates in patients with AD up to 50 times higher compared to those in the control group [34,35]. The caspase family of proteins is a primary apoptosis effector, among which cleaved caspase-3 is a marker of early apoptosis [36,37,38]. Therefore, we measured the expression of cleaved caspase-3 both in vivo and in vitro. The results showed that the levels of cleaved caspase-3 were significantly higher in the hippocampi of mice infected with *T. denticola* and in N2a cells stimulated with *T. denticola*. These findings indicated that *T. denticola* oral infection could induce neuronal apoptosis.

To investigate the potential mechanism by which *T. denticola* oral infection triggered apoptosis in the hippocampi of mice, we further looked at the mRNA levels of *Bcl2* and its family members, *Jnk2*, *Jnk3*, and *Smac*, which are important in regulating the mitochondrial pathway in apoptosis. The BCL-2 protein family regulates cellular life and death signals and mediates intrinsic apoptotic pathways [39]. Studies have reported that aberrant expression of BCL-2 family proteins is closely related to AD [40]. Pro-survival (e.g., BCL-2, BCL-XL, BCL-W) and pro-death (e.g., BAK, BAX, BID, BIM, BIK) proteins are members of the BCL-2 family. We found that *Bcl-2* and *Bclw* were down-regulated and *Bax* and *Bim* were up-regulated, with *Bclw* showing the most significant change. BCL-W is an anti-apoptotic protein that protects against AD pathology in vivo and in vitro [41]. In addition, BCL-W has been shown to protect neurons from Aβ-induced neuronal apoptosis by inhibiting the mitochondrial release of SMAC [42]. Consistent with these findings, our study provided further evidence of the role of BCL-W in neuronal apoptosis. 

SAPK/JNK, a distant member of the mitogen-activated protein kinase (MAPK) superfamily, has previously been shown to trigger neuronal degeneration and death in different brain pathological conditions and diseases [43,44]. JNK has three isoforms (JNK1, 2, and 3) that can play different roles in the regulation of apoptosis [45]. JNK activation has been linked to transcriptional regulation of BCL-2 family members and plays a critical role in neuron death, senile plaque formation, and tau phosphorylation in AD [46,47,48]. Moreover, Yao et al. reported that Aβ reduced BCL-W protein levels via a JNK-dependent mechanism [46]. A previous study found that SMAC/DIABLO, a mitochondrial protein that may promote apoptosis by neutralizing one or more members of the IAP family of apoptosis inhibitory proteins [49]. SMAC was released from the mitochondrial inner membrane space into the cytoplasm to bind and activate caspases under pro-apoptotic conditions and in response to mitochondrial outer membrane permeabilization [50]. However, BCL-W inhibited the release of SMAC from mitochondria and prevented apoptosis [51]. Studies have suggested that downregulation of BCL-W and subsequent SMAC release might be key components in the pathway of Aβ-induced neuronal apoptosis [46]. Consistent with these findings, we discovered that in response to *T. denticola* infection, BCL-W was down-regulated, while SAPK/JNK and SMAC were up-regulated, implying that *T. denticola* may induce neuronal apoptosis via activation of the JNK pathway, down-regulation of the anti-apoptotic protein BCL-W, and release of the pro-apoptotic protein SMAC. However, further studies are needed to address the role of *T. denticola* infection in neuronal apoptosis in AD.

Aβ aggregation is a key step in the pathological process of AD. Previous research has shown that Aβ accumulated primarily extracellularly in brain tissue [52]. An increasing number of studies have found evidence for intracellular accumulation of Aβ [53,54,55]. In our previous study, we found that *T. denticola* can enter the mouse hippocampus and directly induce intra- and extracellular Aβ_1–40_ and Aβ_1–42_ accumulation in the hippocampus [20]. As Aβ is known to be neurotoxic and may mediate neuronal apoptosis through endogenous pathways, we hypothesized that *T. denticola* infection and Aβ might promote neuronal apoptosis through similar mechanisms. In the present study, Aβ_1__–42_- and *T. denticola*-treated N2a cells showed increased apoptosis, up-regulation of cleaved caspase-3, SAPK/JNK, and SMAC, and down-regulation of BCL-W. Moreover, an Aβ inhibitor reversed the *T. denticola*-induced N2a cell apoptosis. KMI1303 is an inhibitor of β-secretase known to inhibit *T. denticola*-inducing Aβ production. In our previous study, we confirmed that the expression of Aβ_1–42_ in the KMI1303-treated group is significantly lower than that in the coculture group [20]. These results suggested that Aβ_1__–42_ was involved in the effect of *T. denticola* infection on neuronal apoptosis. Aβ aggregation was found to be the initiating factor in neuronal degeneration in mice, and intracerebral injection of Aβ_1__–42_ induced neuronal damage and caspase cleavage in the hippocampi of rats [56,57,58]. Mitochondrial apoptotic pathways that trigger mitochondrial dysfunction and DNA damage in Aβ-exposed cells are likely involved in the Aβ neuronal toxicity cascade [59]. Furthermore, Aβ induced apoptosis by modulating the expression of apoptosis-related genes, such as the BCL-2 family of proteins [60]. Longpre et al. observed JNK activation in Aβ-treated neurons and inhibition of JNK activation significantly attenuated Aβ-induced neuronal toxicity [61]. These results suggested that Aβ_1–42_ might mediate the effect of *T. denticola* on neuronal apoptosis by reducing BCL-W expression via JNK activation, resulting in SMAC release (Figure 5).

It is worth noting that previous studies have provided evidences that *P. gingivalis* plays a critical role in the pathogenesis of AD by inducing Aβ accumulation, Tau hyperphosphorylation, neuronal apoptosis, and neuroinflammation in mice [13,14,15,16]. Therefore, we set *P. gingivalis*-infection group as positive control, and the results showed that *T. denticola-* and *P. gingivalis*-infection groups had no significant difference in neuronal apoptosis, thus we did not set a *P. gingivalis*-infection in vitro experiments and mainly focused on the impact and mechanism of *T. denticola* infection in AD.

Further studies are needed to determine how oral *T. denticola* treatment could induce intracranial infection and neuro-amyloidosis. Several studies have revealed the relationship between chronic periodontitis or periodontal pathogens and blood-brain barrier (BBB) damage [62,63,64]. Pathogens and/or their virulence factors entering the brain facilitate BBB disruption in AD, which may be an early feature of the disease [65]. Future research needs to explore whether *T. denticola* infection disrupts the integrity of the hippocampal BBB and whether virulence factors of *T. denticola* can enter the hippocampus and play a role in the pathogenesis of AD.

## 4. Materials and Methods

### 4.1. Bacterial Strains and Culture Conditions 

*T. denticola* ATCC 35405 was cultured in a new oral spirochete (NOS) medium in an anaerobic system as described previously [20]. *P. gingivalis* ATCC 33277 was cultured in a brain-heart infusion medium (OXOID, Basingstoke, UK) supplemented with defibrinated sheep blood, hemin (0.5 mg/mL), and menadione (10 mg/mL) in an anaerobic system (Gene Science, Cambridge, MA, USA) [15]. Bacterial were collected and washed twice with PBS. Bacterial concentrations were measured using a spectrophotometer at an optical density of 600 nm, then diluted with PBS containing 3% carboxymethyl cellulose (CMC) to a concentration of 10^9^ CFU/mL.

### 4.2. Mouse Treatment

All animal experiments were conducted at the State Key Laboratory of Oral Diseases and were approved by the Research Ethics Committee of West China Hospital of Stomatology (WCHSIRB-D-2019-013). Fifteen 8-week-old male C57BL/6 mice (20–22 g) were purchased from the Animal Experiment Center of Sichuan University. All the animal studies were reported following the ARRIVE guidelines. Mice were housed in individually ventilated cages in a standard environment (24–26 °C room temperature, 55% ± 10% humidity) under specific-pathogen-free conditions on a 12-h light/dark cycle with free access to water and food. They were randomly divided into three groups: sham-infection (blank control), *T. denticola*-infection (experimental group), and *P. gingivalis*-infection (positive control). (n = 5 in each group). G*Power 3.1 software (Düsseldorf, Germany) was used to calculate the sample size [66], which was based on the data from our previous study. According to the difference between two independent groups (*t*-test), the sample size was calculated based on the expression levels of Aβ_1–40_ in the hippocampi of two groups (Mean = 305.2, SD = 7.395/Mean = 553.8, SD = 100.2) with an alpha level of 0.05 (type II error) and a power of 95% (type I error). The sample size was calculated to be 4. Considering the possibility of unexpected death during the experiment, 5 mice were included in each group. 

To suppress endogenous oral microorganisms, all mice were given 1 mg/mL of kanamycin in drinking water for 3 days before the first oral administration of periodontal bacteria. The experimental and positive control groups were orally administered the appropriate bacterium (10^9^ CFU/mL/50 µL) for 24 weeks at a frequency of three times per week [14,67], while the blank control group received an equal volume of PBS with 3% CMC solution. The mice were anesthetized with pentobarbital sodium euthanasia one week after the final treatment, and their hearts were quickly perfused with chilled PBS (0.1 M, pH 7.3) (Appendix A). The hippocampi were dissected, and the left hemispheres were fixed with 4% paraformaldehyde and the right hemispheres were stored at −80 °C. 

### 4.3. PCR

A DNeasy Blood & Tissue Kit (Qiagen, Los Angeles, CA, USA) was used to extract genomic DNA from saliva. DNA amplification was performed with a PCR amplification kit (Takara, Tokyo, Japan). Briefly, the PCR mixture was made up of 12.5 μL Taq PCR Master Mix, 100 ng (100 ng/μL) DNA sample, 1 μL forward primer, 1 μL reverse primer, and 9.5 μL sterilized ddH_2_O. The reaction was carried out at 94 °C for 4 min, 40 cycles of 94 °C for 30 s, 55 °C for 5 s and 72 °C for 30 s/60 s; and 72 °C for 2 min. The amplified DNA products were electrophoresed on a 2% agarose gel at 100 V for 30 min. To determine whether the samples contained *T. denticola* ATCC 35405 or *P. gingivalis* ATCC 33277, the specific bands of the samples were compared to those in the positive group. The PCR primer sequences were as follows: (1) *T. denticola*: forward, 5′-TAATACCGAATGTGCTCATTTACAT-3′; reverse, 5′-CTGCCATATCTCTATGTCATTGCTCTT-3′; product, 860 bp; (2) *P. gingivalis*: forward, 5′-AGGCAGCTTGCCATACTGCG-3′; reverse, 5′-ACTGTTAGCAACTACCGATGT-3′; product, 405 bp (Sangon Biotech Co., Ltd., Shanghai, China).

### 4.4. Measurement of Alveolar Bone Loss

The maxilla was extracted and fixed in 4% paraformaldehyde for 48 h. Scanning was done with a Micro-CT instrument (Scanco Medical, Zurich, Switzerland) under the following conditions: samples were placed to make the long axis of the tooth parallel to the scanning ray, current 145 mA, voltage 55 kVp, resolution 12 μm, integration time 300 ms. Materialise Mimics software was used to perform the three-dimensional reconstruction. The distance from the CEJ to the ABC of the second maxillary molar was measured at three different sites (the distal root, root furcation groove, and mesial root) on the lingual side with ImageJ software. The alveolar bone level of the maxilla was calculated by averaging the values from these three sites.

### 4.5. Cell Culture and Treatment 

N2a cells were cultured in Dulbecco’s modified Eagle’s medium (DMEM) supplemented with 10% fetal bovine serum in a humidified (5% CO_2_, 37 °C) incubator. Following confirmation of cell status, cells were plated in 6-well or 24-well tissue culture plates. The experimental groups were co-incubated with *T. denticola* (multiplicity of infection (MOI) 1:100) or 25 μM Aβ_1–40_, 25 μM Aβ_1–42_ for 2 h, while the control group was incubated with DMEM containing 10% FBS. The medium was then replaced with an equal volume of fresh medium, and the cells were cultured at 37 °C for 12 h (Appendix A).

To investigate the role of Aβ in *T. denticola*-induced apoptosis, N2a cells in the experimental groups were pretreated with 1 μM KMI1303 (Bioss, Beijing, China) for 4 h before being incubated with the *T. denticola* for 2 h. The medium was then replaced with an equal volume of fresh medium, and the cells were cultured at 37 °C for 12 h (Appendix A). 

### 4.6. TUNEL Staining

TUNEL staining was carried out as described previously with some modifications using a TUNEL kit (Roche, Basel, Switzerland) [68]. Briefly, the left hippocampus was paraffinized and sectioned, then deparaffinized in xylene, rehydrated using an ethanol gradient (100%, 95%, 90%, 80%, and 70%; 3 min per rehydration step), washed in PBS (5 min), and incubated with 20 μg/mL proteinase K in 10 mM Tris-HCl, pH 7.4–7.8 at 37 °C for 20 min in a humidified chamber. TUNEL staining was used to detect DNA fragmentation [69]. The slides were incubated with a TUNEL reaction mixture in a humidified chamber at 37 °C for 60 min before being rinsed with PBS three times. The slides were incubated with converter-POD in a humidified chamber at 37 °C for 30 min, then with PBS three times. One hundred microliters of DAB (5 μL 20× diaminobenzidine 3 + 1 μL 30% H_2_O_2_ + 94 μL PBS) substrate was added to the slides and allowed to react at 15–25 °C for 10 min. The slides were rinsed with PBS and counterstained with hematoxylin. After washing with running water, the slides were dehydrated, cleared, and sealed with neutral balsam. Images were captured using a fluorescence microscope.

### 4.7. Western Blotting

Western blotting was performed as described in a previous study with some modifications [20]. Protein extracts from cells or hippocampi were prepared in a modified RIPA buffer supplemented with protease inhibitors (200612, Signalway Antibody, Greenbelt, MD, USA). The BCA method was used to determine the concentration of protein. The protein extracts were boiled after being diluted in SDS-PAGE protein loading buffer (5×) (Beyotime, Shanghai, China) at a ratio of 4:1. Following separation, the proteins were transferred to polyvinylidene difluoride membranes and blocked in TBST buffer (20 mM Tris–HCl, pH 7.4, 137 mM NaCl, and 0.1% Tween-20) with 5% non-fat milk at 37 °C for 1 h, and incubated at 4 °C with primary rabbit polyclonal antibodies (cleaved caspase-3, 49500, 1:500, Signalway Antibody; BCL-W, 40641, 1:1000, Signalway Antibody; SAPK/JNK (pThr183), 11249, 1:500, Signalway Antibody; SMAC/DIABLO, 39330, 1:500, Signalway Antibody; GAPDH, 21612, 1:3000, Signalway Antibody) overnight. After extensive rinsing, the membranes were incubated with the appropriate HRP-conjugated secondary antibody, then visualized using Super ECL Plus reagents. The gray values of the protein bands were quantified by the optical density using ImageJ software (1.41v, US National Institutes of Health, Bethesda, MD, USA).

### 4.8. qRT-PCR

The procedure was carried out in accordance with the MIQE guidelines [70]. Total RNA was extracted from mouse hippocampi or cells using an RNApure total RNA fast isolation kit (BioTeke, Beijing, China), then reverse-transcribed using an Evo M-MLV RT kit with gDNA Clean for qPCR II (Accurate Biology, Changsha, China) according to the manufacturer’s instructions. The resulting cDNA served as a template for quantitative PCR analysis using gene-specific primers (TSINGKE, Beijing, China). Real-time quantitative polymerase chain reactions were performed with TB Green Premix Ex Taq II (TAKARA, Tokyo, Japan) using an Applied Biosystems QuantStudio 6 Flex Real-Time PCR System. The cycling conditions were as follows: initial denaturation at 95 °C for 30 s, then 40 cycles of 95 °C for 5 s, and 60 °C for 30 s, followed by 95 °C for 15 s, 60 °C for 1 min, then 95 °C for 15 s. The fluorescence intensity was monitored at the end of each amplification step. Quantitative measurements of the target gene levels were normalized to GAPDH and the results were expressed as fold changes of the threshold cycle (Ct) value relative to control using the 2^−ΔΔCt^ method. Primers and amplicon size were shown in Table 1.

### 4.9. Statistical Analysis

Data were presented as the mean ± standard deviation (SD) and analyzed using SPSS 16.0 statistical software (SPSS Inc., Chicago, IL, USA). The student’s *t*-test was used to analyze statistical differences. Differences were considered significantly different if the *p*-value was <0.05.

## 5. Conclusions

In conclusion, this study showed that *T. denticola* oral infection could induce alveolar bone loss and neuronal apoptosis in mice. The potential mechanism might be related to the intrinsic mitochondrial pathway mediated by Aβ. These findings provided novel insights into the important role of *T. denticola* in AD pathogenesis and suggested that prevention and treatment of periodontitis may be beneficial for preventing and slowing the progression of AD.

## Figures and Tables

**Figure 1 pathogens-11-01150-f001:**
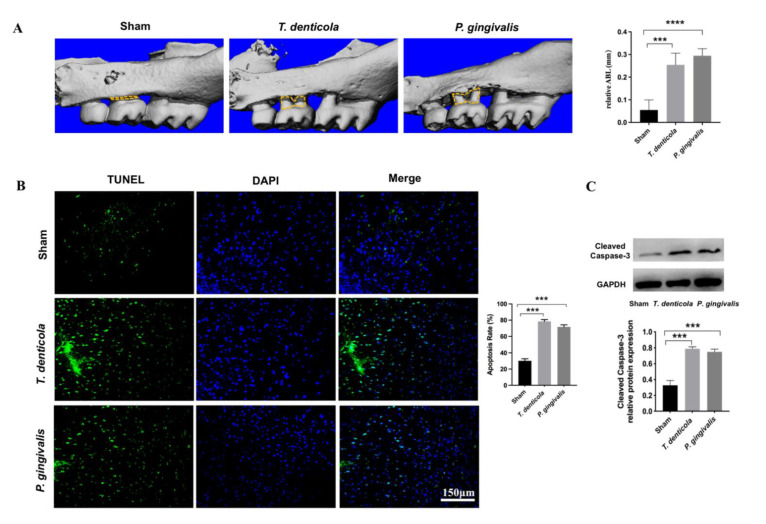
Oral infection with *T. denticola* induced alveolar bone resorption and neuronal apoptosis in the hippocampi of mice. (**A**) Morphometric evaluation of total horizontal ABL in mice. The yellow dotted line indicates the area between the CEJ of the maxillary molar and the ABC; (**B**) The apoptosis rate in the hippocampi of mice was examined using TUNEL staining, *n* = 4 mice per group. (**C**) The protein levels of cleaved caspase-3 in the mouse hippocampus were examined by western blotting, *n* = 3 mice per group. Results are presented as the mean ± SD, ***: *p* < 0.001, ****: *p* < 0.0001.

**Figure 2 pathogens-11-01150-f002:**
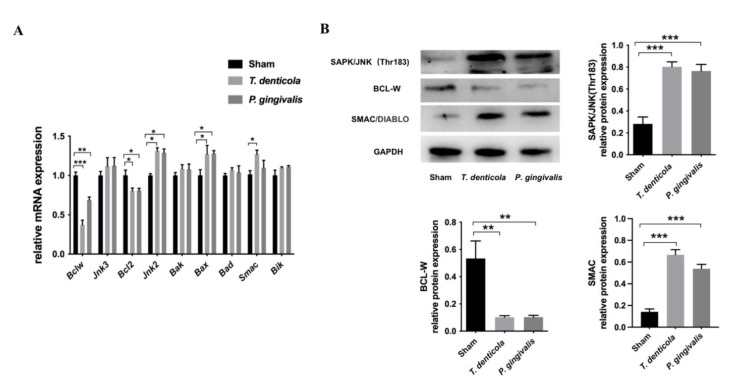
*T. denticola* oral infection regulated the expression levels of apoptosis-associated genes and proteins. (**A**) The gene expression levels of *Bclw*, *Bcl2*, *Bik*, *Bax*, *Bak*, *Bad*, *Jnk2*, *Jnk3*, and *Smac* in the hippocampi of mice were measured using Quantitative real-time PCR (qRT-PCR). The values are shown as the mean ± SD of three independent experiments, *n* = 4 mice per group. (**B**) The protein levels of BCL-W, SAPK/JNK, and SMAC in the mice hippocampi were examined by western blotting, and the results were quantitatively analyzed. Results are presented as the mean ± SD, *n* = 3 mice per group *: *p* < 0.05, **: *p* < 0.01, ***: *p* < 0.001.

**Figure 3 pathogens-11-01150-f003:**
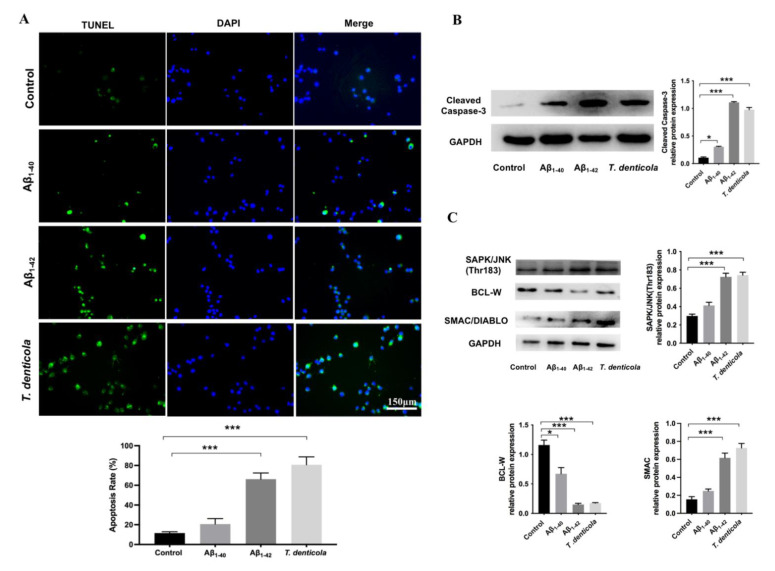
*T. denticola* and Aβ_1–42_ induced apoptosis in N2a cells via a similar mechanism. (**A**) N2a cells were co-cultured with *T. denticola*, Aβ_1–40_, and Aβ_1–42_, and the apoptosis rate was examined using TUNEL staining. (**B**) The protein levels of cleaved caspase-3 in N2a cells in each group were examined by western blotting, and the results were quantitatively analyzed. (**C**) The protein levels of SAPK/JNK, BCL-W, and SMAC in N2a cells in each group were examined by western blotting, and the results were quantitatively analyzed. Results are presented as the mean ± SD, *: *p* < 0.05, ***: *p* < 0.001, *n* = 3 per experiment.

**Figure 4 pathogens-11-01150-f004:**
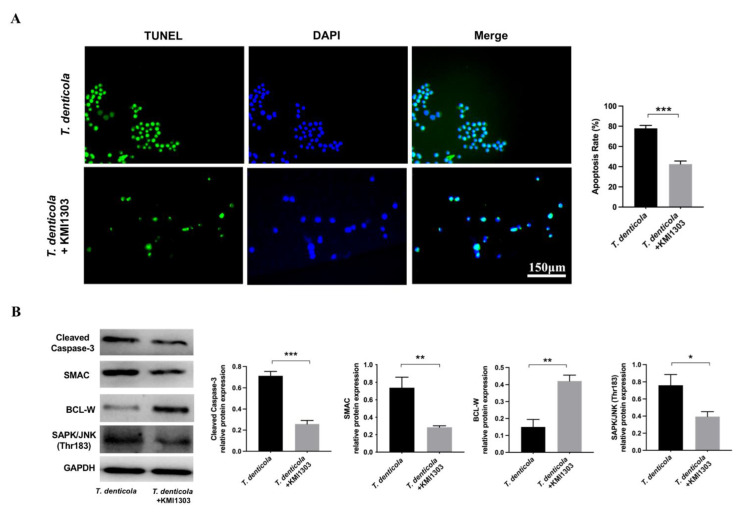
Amyloid-β mediated the effect of *T. denticola* infection on neuronal apoptosis. (**A**) N2a cells were pretreated with KMI1303, then co-cultured with *T. denticola*, and the apoptosis rate was examined using TUNEL staining. (**B**) The protein levels of cleaved caspase-3, BCL-W, SAPK/JNK, and SMAC in N2a cells in each group were assessed by western blotting, and the results were quantitatively analyzed. Results are presented as the mean ± SD, *: *p* < 0.05, **: *p* < 0.01, ***: *p* < 0.001, *n* = 3 per experiment.

**Figure 5 pathogens-11-01150-f005:**
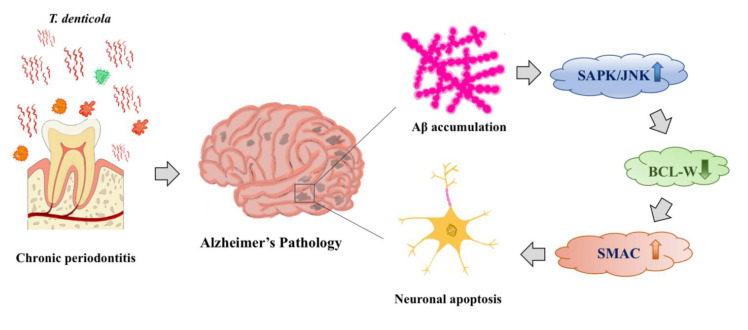
Schematic overview of *T. denticola*-induced neuronal apoptosis. *T. denticola* infection induces Aβ accumulation in hippocampi, Aβ induces neuronal apoptosis via activation of the JNK pathway, down-regulation of the anti-apoptotic protein BCL-W, and release of the pro-apoptotic protein SMAC.

**Table 1 pathogens-11-01150-t001:** Specific primer pairs used in qRT-PCR.

Gene Name	Forward Primer (5′-3′)	Reverse Primer (5′-3′)	Amplicon Size	Genebank Accession Number
*Bcl-2*	GAGAGCGTCAACAGGGAGATG	CCAGCCTCCGTTATCCTGGA	108 bp	AH001858
*Bcl-w*	ACTGAACAGGGTTTTGTGACTT	CCAGTTATTCCCCTTAGCAAGGT	105 bp	U59746
*Bax*	CCGGCGAATTGGAGATGAACT	CCAGCCCATGATGGTTCTGAT	229 bp	AB029557
*Bak*	CAGCTTGCTCTCATCGGAGAT	GGTGAAGAGTTCGTAGGCATTC	108 bp	Y13231
*Bad*	TGAGCCGAGTGAGCAGGAA	GCCTCCATGATGACTGTTGGT	154 bp	NM_001285453
*Bik*	ACGTGGACCTCATGGAGTG	TGTGTATAGCAATCCCAGGCA	129 bp	NM_007546
*Smac*	TCTTGGCTAACTCTAAGAAACGC	TGCTTCGTTACTGAGAGACTGA	140 bp	NM_023232
*Jnk2*	AGAACCAAACGCACGCAAAG	GCTGAATGCAGATGCTTGATG	250 bp	AB005664
*Jnk3*	CCATGTCTGTGTTCTTTCTCACG	TTGGTTCCAACTGTGAAGAGTC	118 bp	AB005665
*Gapdh*	TGGCCTTCCGTGTTCCTAC	GAGTTGCTGTTGAAGTCGCA	178 bp	AY618199

## Data Availability

The data presented in this study are available on request from the corresponding author (X.C. and H.W.).

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
