# Peer review of "Treponema denticola* Induces Neuronal Apoptosis by Promoting Amyloid-β Accumulation in Mice"

_pathogens, 2022, doi:10.3390/pathogens11101150_

Round 1

Reviewer 1 Report

This manuscript by Linrui Wu et al. describes the effect of periodontal pathogen Treponema denticola on neuronal apoptosis using in vivo and in vitro model. Overall, the manuscript was well-written and the conclusion was well supported by the data. I request a minor modification as following: 

Animal study shows that both T. denticola and P. gingivalis can induce neuronal apoptosis in hippocampus, but following in vitro experiments were conducted only with T. denticola. I think it would be better if the in vitro data with P. gingivalis are provided together. Alternatively, I recommend to add a brief description why the in vitro experiments were done with T. denticola only.

Author Response

Point 1: Animal study shows that both T. denticola and P. gingivalis can induce neuronal apoptosis in hippocampus, but following in vitro experiments were conducted only with T. denticola. I think it would be better if the in vitro data with P. gingivalis are provided together. Alternatively, I recommend to add a brief description why the in vitro experiments were done with T. denticola only.

Response: We thank the reviewer for this constructive suggestion. In the present study, we focused on the role of T. denticola infection on the pathogenesis of AD. Previous studies have provided evidence that P. gingivalis plays a critical role in the pathogenesis of AD. P. gingivalis infection can induce Aβ accumulation, Tau hyperphosphorylation, neuronal apoptosis, and neuroinflammation in mice. Therefore, we set the P. gingivalis-infection group as a positive control, and the results showed that T. denticola- and P. gingivalis-infection groups had no significant difference in neuronal apoptosis, thus we did not set a P. gingivalis-infection in vitro experiments and mainly focused on the impact and mechanism of T. denticola infection. We have clarified this in the revised manuscript.

Reviewer 2 Report

Dear authors,

This article reports in vivo experiments realized in mice that aimed to investigate whether oral infection with T. denticola induced alveolar bone loss and neuronal apoptosis. The methodology is well described and robust. The authors used several methods such as Micro-CT was employed to assess the alveolar bone resorption, Western blotting, quantitative PCR, and TUNEL staining to detect the apoptosis-associated changes in mouse hippocampi. Moreover, N2a were co-cultured with T. denticola to verify in vivo results. The authors conclude that oral infected T. denticola caused alveolar bone loss, and induced neuronal apoptosis by promoting Aβaccumulation in mice, providing evidence for the link between periodontitis and AD.

Just some minor comments

Introduction

1.     Replace AD the first time by Alzheimer disease

Results

1.     Write in italic the name of bacteria

Author Response

Point 1: Introduction   Replace AD the first time by Alzheimer disease.

Response 1: We have revised the manuscript accordingly.

Point 2: Results  Write in italic the name of bacteria.

Response 2: We have checked all the names of bacteria in the manuscript and revised them accordingly.

Reviewer 3 Report

Previous published research from this group demonstrated that oral infections with T. denticola could promote AD pathology in the hippocampi of mice, including an increase in Aβ burden, tau hyperphosphorylation, and neuroinflammation. In this study they hypothesized that T. denticola infection might induce neuronal apoptosis and accelerate the pathological progression of AD by increasing the Aβ burden. The group induced well documented periodontitis with Tdenticola and Pgingivalis infection and measured neuronal apoptosis by Tunel and cleavage caspase 3 by Western blot.

Results:

Fig 1 B. Image brightness of TUNEL and DAPI was difficult to see in relatively low-resolution image

Author Response

Point 1: Fig 1B. Image brightness of TUNEL and DAPI was difficult to see in relatively low-resolution image.

Response 1: As suggested by the reviewer, we have replaced them with high-resolution images and increased the brightness of the images in the revised manuscript.
